

# Soil organic carbon estimation using remote sensing data-driven machine learning

Qi Chen, Yiting Wang and Xicun Zhu

College of Resources and Environment, Shandong Agricultural University, Taian, China

## ABSTRACT

Soil organic carbon (SOC) is a crucial component of the global carbon cycle, playing a significant role in ecosystem health and carbon balance. In this study, we focused on assessing the surface SOC content in Shandong Province based on land use types, and explored its spatial distribution pattern and influencing factors. Machine learning methods including random forest (RF), extreme gradient boosting (XGBoost), and support vector machine (SVM) were employed to estimate the surface SOC content in Shandong Province using diverse data sources like sample data, remote sensing data, socio-economic data, soil texture data, topographic data, and meteorological data. The results revealed that the SOC content in Shandong Province was 8.78 g/kg, exhibiting significant variation across different regions. Comparing the model error and correlation coefficient, the XGBoost model showed the highest prediction accuracy, with a coefficient of determination ($R^2$) of 0.7548, root mean square error (RMSE) of 7.6792, and relative percentage difference (RPD) of 1.1311. Elevation and Clay exhibited the highest explanatory power in clarifying the surface SOC content in Shandong Province, contributing 21.74% and 13.47%, respectively. The spatial distribution analysis revealed that SOC content was higher in forest-covered mountainous regions compared to cropland-covered plains and coastal areas. In conclusion, these findings offer valuable scientific insights for land use planning and SOC conservation.

## INTRODUCTION

Soil is considered the primary and largest terrestrial carbon reservoir, with a global carbon stock of 1,550 Pg (*Lal et al., 2018*), containing carbon contents approximately two to three times higher than those found in the atmosphere and vegetation (*Scharlemann et al., 2014*). Even minor changes in terrestrial carbon pools could have significant implications for climate change and global warming (*Lin et al., 2023*). Organic carbon stored in soils below a depth of 30 cm represents an average of 33% of the total SOC stock (*Lacoste et al., 2014*). The storage of SOC is a critical function of soils, influencing climate regulation and other soil functions (*Wiesmeier et al., 2019*). The increase in SOC content can enhance soil structure, improve soil water retention and fertility, stimulate plant growth, and enhance the diversity and activity of soil microorganisms (*Lehmann & Kleber, 2015*). Therefore,

Corresponding author
Xicun Zhu, zhuxcsdau@163.com

acquiring information on SOC content and spatial variations is crucial for enhancing soil structure, ensuring food security, and mitigating global climate change (*Ren et al., 2021*).

Conventional soil mapping methods face challenges in the collection and analysis of a large number of sampling points (*Ren et al., 2021*). Remote sensing techniques can effectively monitor the spatial and temporal dynamics of SOC, potentially enhancing predictions by utilizing ancillary variables, scale-specific methods, and improved model integration (*Croft, Kuhn & Anderson, 2012*). For instance, in global SOC and Chinese soil attribute mapping, environmental variables and multispectral remote sensing data are utilized (*Liu et al., 2022*; *Hengl et al., 2017*). Hyperspectral inversion methods, based on spectral index models, are widely employed for estimating SOC content (*Wei et al., 2020*; *Zhao et al., 2022*). Studies have also utilized Gaofen-5 (GF-5) hyperspectral remote sensing images in conjunction with the partial least squares method to develop models for soil sanding index, soil degradation index, normalized brightness index, and soil salinity index for predicting surface soil organic matter content (*Zhao, Cui & Liu, 2020*).

Different models also influence the estimation accuracy of remote sensing retrieval of SOC. Machine learning (ML) algorithms have significant advantages in data processing, model construction, and optimization. They allow for better handling of issues such as model uncertainty and nonlinearity. These algorithms include support vector machine (SVM), random forest (RF), extreme gradient boosting (XGBoost), and Cubist, *etc*. *Taghizadeh-Mehrjardi et al. (2020)* compared the accuracy of six machine learning algorithms in SOC estimation and found that deep learning neural networks (DNN), RF, and XGBoost achieved higher accuracy than SVM, artificial neural network (ANN), and Cubist. In a previous study (*Wang et al., 2023*), the RF model showed the highest simulation accuracy for forest areas, while the XGBoost model performed best for farmland areas. In a study by *Yuan et al. (2021)*, two combinations of environmental variables were used as input datasets to simulate the prediction and accuracy of SOC content in the surface layer of arable land. The study employed the RF algorithm and compared it with the Ordinary Kriging (OK) interpolation model. *Meliho et al. (2023)* used four ML algorithms to predict SOC stock (SOCS) in the Ourika Basin of Morocco: Cubist, RF, SVM, and Gradient Boosting Mechanism (GBM). The results showed that Cubist ($R^2$ = 0.86, RMSE = 11.62 t/ha) and RF ($R^2$ = 0.79, RMSE = 13.26 t/ha) had the highest predictive ability for SOCS

In addition, some research has developed supervised machine learning methods for predicting marine biochemical processes and acquired advantages in accuracy and effectiveness (*Adhikary et al., 2024*), and Gradient Boosting Regression had most effective with an $R^2$ of 0.904 and MSE of 0.0001. *Adhikary et al. (2021)* also applied various machine learning regression algorithms (random forest, extra trees, bagged, and gradient boosted regressors) to forecast phytoplankton levels globally, which found that the extra tree regression model performed the best, with an $R^2$ of 0.96. Machine learning and deep learning techniques were used to study the relationship between marine chlorophyll and physicochemical features, and found that random forests performed the best among all features, with a classification accuracy of 93.92% (*Tiwari, Adhikary & Banerjee, 2022*).

**Table 1 Main references comparison table.**

| Author | Dataset used | Advantages | Disadvantages | Model used | Results |
|---|---|---|---|---|---|
| Hengl et al. (2017) | 150,000 soil profiles, MODIS land products, Climate | 158 remote sensing-based soil covariates. Procuct SoilGrids 250 m data | Time matching between soil sampling data with remote sensing data is poor. Lack human activities variables | RF and gradient boosting and/or multinomial logistic regression | All soil sampling data was used to build soil properties prediction. No accuracy validation |
| Taghizadeh-Mehrjardi et al. (2020) | 154 soil profiles from arid and 99 from semi-humid areas, SRTM DEM, Landsat-8 and Sentinel-2 images | Model stacking, improves prediction accuracy, effectively handles complex multivariate datasets | Model input data lacks climate, soil properties, and socio-economic data | LSTM, RF, ANN, XGBoost, AvNNet, DNN | RMSE values after model stacking are 17% and 9% in arid and semi-humid areas, lower than the best individual models |
| Ye et al. (2021) | 295 soil samples, GF-6, Landsat8 images, SRTM image data | GF-6 multispectral satellite data suitable for SOM retrieval, improves precision agriculture policy-making | Lacks climate and human activity variables | RF, LightGBM, GBDT, XGBoost | XGBoost model using feature-optimized dataset performed better than other models with $R^2 = 0.771$ |
| Yuan et al. (2021) | 1,257 SOC measured data points, Landsat8 OLI images, ASTER GDEM, 22 climate stations data | Based on a multivariate combination (remote sensing + climate + soil properties), improves SOC prediction accuracy | Potential dataset errors, model applicability limitations | RF algorithm | The RF model using all variables performed best, and its model prediction accuracy significantly improved compared with the model without soil attributes (R increased by 7.95%, RMSE decreased by 45.13%) |
| Liu et al. (2022) | 5,000 soil profiles during 2010 and 2018, Landsat, SRTM DEM, climate | The time matching between remote sensing data and soil data is good. | Lack human activities variables | Quantile regression forest | $R^2$ ranged from 0.36 to 0.71. Product 90 m spatial resolution SOC mapping of China. |
| Nguyen et al. (2022) | Field survey soil samples from Western Australia, binary land use map generated from 266 digitized points | Optical and SAR data fusion, advanced machine learning techniques, improves SOC estimation robustness | Only uses remote sensing data, lacks environmental covariates | XGBoost, RF, SVM | XGBoost obtained higher estimation effective than RF and SVM ($R^2 = 0.870$, RMSE = 1.818 tC/ha) |
| Meliho et al. (2023) | 420 soil samples, soil properties, climate, terrain, and remote sensing | Multiple environmental covariates improve SOC prediction accuracy | Complex methodology, choice of modeling predictor variables may affect performance | Cubist, RF, SVM, Gradient Boosting Mechanism | RF ($R^2 = 0.79$, RMSE = 1.2) and Cubist ($R^2 = 0.77$, RMSE = 1.2) obtained higher retrieval accuracy than other models |

In addition, soil type and vegetation cover type also have a significant impact on SOC sequestration. As shown in previous studies, soil mineralization levels are closely related to SOC content, with soils having low mineralization levels typically having higher SOC content, while soils with high mineralization levels tend to have relatively lower SOC content (*Jobbágy & Jackson, 2000*).

Many studies have been conducted both domestically and internationally on remote sensing retrieval of SOC content and spatial distribution using machine learning models combined with covariates. The following table summarizes the datasets used in the

literature, their advantages and disadvantages, the models used, and their results in some previous study (Table 1). According to these previous study, multi-source remote sensing data is suitable for SOC predication in global or region research, and SAR data input can improve prediction accuracy, RF and gradient boosting machine learning model can obtain higher accuracy than other models.

Above all, in this study, Shandong Province was selected as the study area. Multi-source data including optical, SAR, climate, socio-economic data, and soil properties data were used to build SOC prediction model using RF, XGBoost, and SVM model. Moreover, spatial distribution characteristic of SOC wasanalyzed in combination with land use types. This research aims to deepen the understanding of the spatial distribution and variation patterns of SOC, providing a scientific basis for soil quality assessment and sustainable land management.

## MATERIALS AND METHODS

### Study area

Located in the eastern coastal region of China, Shandong Province spans geographic coordinates from 32°18′28″N to 38°23′23″N and 114°19′50″E to 122°43′36″E, covering a total land area of approximately 157,900 km². Shandong Province exhibits a complex and diverse topography, including plains, mountains, hills, and coastal areas. The central part is characterized by high-altitude mountainous areas, while the western and northern regions consist of plain areas formed by the Yellow River's alluvial deposits. Shandong Province falls within the temperate monsoon climate zone, primarily influenced by marine and monsoon climates, leading to notable variations in annual average temperature and precipitation levels among different regions (Fig. 1).

Shandong Province boasts a diverse range of agricultural land types, such as paddy fields, drylands, orchards, forests, and grasslands. The main soil types in Shandong Province include brown soil, cinnamon soil, tidal soil, saline soil, among others. Various land use types and conversions between farmland, forests, grasslands, buildings, and unused land significantly influence the accumulation and stability of SOC. Agricultural management practices, including fertilization, tillage, and irrigation, can impact the fluctuations in SOC content. Furthermore, socio-economic factors like fertilizer and pesticide usage, land ownership patterns, and rural economic development levels may affect the distribution and variability of SOC content.

### Data resources

#### Sampling SOC data

The collected data were from the Shandong volume of "China Soil Systematics" published in 2019, totaling 123 and covering the whole Shandong region more evenly (*Zhao, Song & Zhang, 2019*). The acquired SOC content dataset underwent several processing steps to ensure data quality and suitability for analysis. These steps included data cleaning, handling missing values, converting data format, selecting relevant features, and partitioning the dataset for training and evaluation.

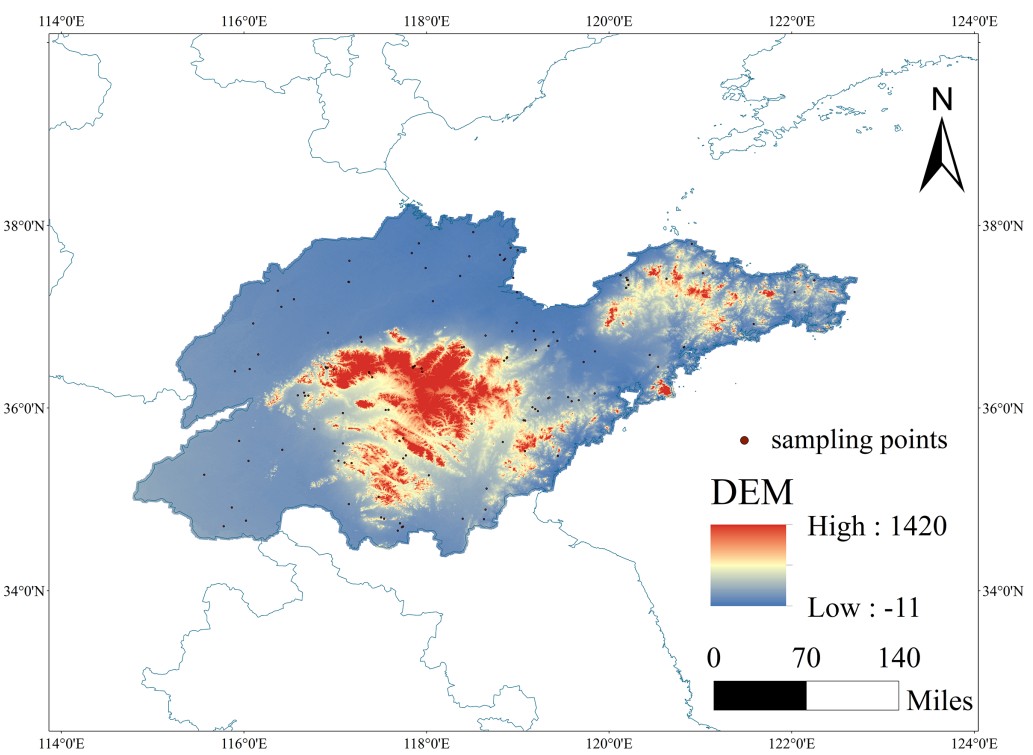

**Figure 1 The spatial location of the study area and the spatial distribution of sampling points.** Map source credit: ALOS PALSAR RTC: ©NASA (2015), ©JAXA, METI (2015), DOI: 10.12078/ 2023010103.

### Acquisition and processing of remote sensing data

The Sentinel-1A VV, VH polarization data, Normalized Difference Vegetation Index (NDVI) data, and Enhanced Vegetation Index (EVI) data for 2015 were obtained from the Google Earth Engine (GEE) platform (*Google Earth Engine Team, 2015*). In GEE, we apply speckle filtering to process the VV and VH polarization data of Sentinel-1A. The Sentinel-1A VV and VH polarization data are subjected to speckle filtering with a window size of 3 × 3 pixels. The blue band (B2), red band (B4), and near-infrared band (B8) of Sentinel-2 are selected to calculate NDVI and EVI. And a 30-day median composite image was generated to reduce the impact of noise and outliers, with a focus on capturing seasonal changes in vegetation indices.

The spatial reference is uniformly converted to WGS 1984, with an Albers projection and a spatial resolution of 1,000 m.

This study combines optical images and radar data to provide a richer set of surface information with different scales and features. This integration improves the accuracy and reliability of estimating SOC content.

### Other geographical data

In this study, environmental covariates were used to build models. Advanced Land Observing Satellite (ALOS) DEM (*Laurencelle, Logan & Gens, 2015*) of ALOS DEM. We first verify the integrity of the data by cross referencing the original metadata provided

by *Laurencelle, Logan & Gens (2015)*. Then, we clean up the data by removing any errors that may affect model performance. The spatial resolution meets the requirements of our research; However, if there are differences, we use bilinear interpolation to adjust the resolution to match our 1,000 m target scale.

The annual mean temperature and precipitation data were obtained from the Resources and Environmental Science and Data Center of the Chinese Academy of Sciences (RESDC) (http://www.resdc.cn) with 1 km resolution. We use spatial interpolation techniques to estimate missing data points, ensuring the final dataset is complete and spatially coherent.

Soil texture data were obtained from National Earth System Science Data Center (http://soil.geodata.cn/data/). We use a standard soil classification system to classify the data and convert units as necessary to maintain consistency in the dataset.

Land use/cover change data (LUCC) were obtained from the Resource and Environment Science Data Center of the Chinese Academy of Sciences (*Xu et al., 2018*) (http://www.resdc.cn/), which included six primary categories: cropland, forestland, grassland, water bodies, urban and rural residential and industrial land, and unused land. We convert classified land use data into a format suitable for integration with other datasets, ensuring that the six main land use categories are clearly defined and mutually exclusive.

Population density data utilized in this study were obtained from the Center for Resource and Environmental Science and Data (https://www.resdc.cn/), providing national population distribution data at a spatial resolution of 1,000 m. We use area weighted interpolation to resample population density data from the original resolution to match our target resolution of 1,000 m. This process ensures that population data is spatially consistent with other environmental covariates.

The basic map used in this study comes from the Resource and Environmental Science Data Registration and Publishing System (RESDC). The specific dataset used is "China's multi-year provincial administrative boundary data" (*Xu, 2023*).

## Screening of environmental factors

There are many environmental factors that affect SOC content, so it is important to screen the model variables before training the model. Since machine models are often referred to as "black boxes" that do not directly reveal the functional relationship between environmental factors and target variables, each variable needs to be eliminated one by one in order to determine its effect on the model. Variables were screened by increasing or decreasing the RMSE, retaining the variable when the RMSE increased, and excluding it when the RMSE increased.

## Retrieval model of SOC

In this study, SVM, RF, and XGBoost machine learning models were selected to retravel SOC content. On the one hand, these machine learning models can effectively handle multidimensional datasets and Collinearity problem. On the other hand, these models were widely used in SOC retrieval and obtained good simulation effect (*Hengl et al., 2017*; *Liu et al., 2022*; *Zhou et al., 2023*). For example, SVM exhibits significant advantages in

addressing small sample problems. It maintains robust classification performance even with limited sample sizes by constructing an optimal hyperplane to separate different data categories (*Yao, 2022*). RF enhances prediction accuracy and robustness by integrating multiple decision trees (*Momade et al., 2020*), while XGBoost supports various objective functions and evaluation metrics, enabling outstanding performance in diverse prediction tasks (*Sagi & Rokach, 2021*). Although studies indicate that deep neural networks (DNN) are more accurate than RF and SVM in SOC simulations in some areas, neural networks require extensive parameter settings and more sample data (*Raczko & Zagajewski, 2017*); given that a small sample size may not be optimal for this study, deep learning was not chosen.

In this study, SVM modeling was implemented using the "svm" function in the "e1071" package, with the "importance" parameter set to TRUE to calculate variable importance and the "proximity" parameter set to TRUE to calculate sample proximity.

RF modeling was implemented using the "randomForest" and "caret" packages in the R programming language. The default value of "ntree" was set to 500, and the "expand.grid()" function in the "caret" package is used, which determines the optimal value of the parameter mtry through grid search. Specifically, we performed a grid search on the mtry parameter, traversing a series of possible values and evaluating the impact of each value on model performance. The role of the expand.grid() function is to generate all possible parameter combinations for grid search.

XGBoost incorporates regularization techniques, efficient parallel processing, feature selection, and missing value handling, among other optimization techniques. The "xgboost" library was loaded using the "library(xgboost)" command. In this model, learning rate and depth of tree were gbtree, 0.4 and 7, respectively. In this study, a grid search strategy was used to fine-tune all parameters using the caret package in R software. The dataset was divided into a training set and a test set with a ratio of 7:3.

## Model accuracy validation

In this study, the accuracy and stability of the models were evaluated using three commonly used metrics: coefficient of determination ($R^2$), root mean square error (RMSE), and relative percent difference (RPD). Higher modeling accuracy indicates stronger stability and predictive capability of the model. A value of $R^2$ closer to 1 indicates a higher explained variance of the target variable, indicating better model fit. A lower RMSE value indicates smaller prediction errors and better model fit. As for RPD, its classification is detailed in *Viscarra Rossel, Taylor & McBratney (2007)*.

$$RMSE = \sqrt{\frac{1}{n}\sum_{i=1}^{n}(P_i - M_i)^2} \tag{1}$$

$$R^2 = \left(\frac{\sum_{i=1}^{n}(M_i - \bar{M})(P_i - \bar{P})}{\sqrt{\sum_{i=1}^{n}(M_i - \bar{M})^2}\sqrt{\sum_{i=1}^{n}(P_i - \bar{P})^2}}\right)^2 \tag{2}$$

$$\mathrm{RPD} \ = \frac{\mathrm{SD_O}}{\mathrm{RMSE}}. \tag{3}$$

In the equation, $P_i$ and $M_i$ represent the simulated and measured values of SOC content (g/kg), respectively, while SDO represents the standard deviation of the observed values.

## RESULTS AND DISCUSSION

### Statistical analysis of SOC content

The average SOC content of the soil samples in this study was 8.78 g/kg. According to the national soil nutrient classification standards, it can be classified as a moderate level (6–12 g/kg). The coefficient of variation (CV) of SOC content in the collected samples was 55.75%, which was a moderate level of variation (Table 2).

Meanwhile, SOC content of different vegetation was different as showed in Table 3. Forest had highest SOC content, mean value of SOC was 18.2 g/kg, but also has the maximum CV (88.7%). Reasons may be that the forest covered large area in Shandong Province, the elevation fluctuated greatly and the sampling site was sparse. SOC of Farmland and grassland was lower with a mean value of 8.5, and 8.1 g/kg, and has the moderate CV (belongs to 49.6–54.3%).

### Accuracy comparison analysis

XGBoost demonstrates the highest predictive accuracy in the validation set, with an $R^2$ of 0.7548, RMSE of 7.6792, and RPD of 1.1311 (Table 4). This indicates that the model can account for 75.48% of the variance in the test data. XGBoost also exhibits the highest $R^2$ value of 0.9573 and RPD value of 3.1859 in the training set, suggesting exceptional predictive capability and accuracy. These findings are consistent with previous research results (*Emadi et al., 2020*; *Nguyen et al., 2022*; *Ye et al., 2021*). While all three models perform well on the training set, they show poor performance on the test set or new data. This highlights our limitation, as the insufficient training samples lead to overfitting. Due to the lower generalization error of the RF model, which is more robust to noise (*Breiman, 2001*) and capable of handling non-linear and hierarchical relationships between SOC and predictor variables (*Zhang et al., 2017*), some studies (*Lamichhane, Kumar & Wilson, 2019*) suggest that the RF model can better simulate SOC content. This contrasts with our study, which can be attributed to differences in research fields, feature variable selection, and spatial resolution choices.

### Driving factors of SOC simulation analysis

RF was selected to analyze the explanatory power of each variable in predicting the surface SOC content. RF was chosen for feature variable analysis because it can handle a larger number of variables, improve the accuracy and performance of the classification task, efficiently identify correlated variables (*Chen et al., 2020a*), and generate *p*-values to determine significantly correlated features while controlling the false discovery rate (*Paul & Dupont, 2015*).

Figure 2 illustrates the significance of each independent variable in the analysis. The results show that elevation had the highest explanatory power at 21.74% in explaining

**Table 2 SOC content statistical description.**

| Sample set | Sample size | Maximum value | Minimum value | Mean value | Standard deviation | Coefficient of variation (%) |
|---|---|---|---|---|---|---|
| Whole dataset | 123 | 27.8 | 2.2 | 8.78 | 4.89 | 55.75 |
| Modeling dataset | 86 | 27.8 | 2.2 | 8.63 | 4.07 | 47.19 |
| Validation dataset | 37 | 27.6 | 3.1 | 9.15 | 6.47 | 70.78 |

**Table 3 SOC content statistical description for different vegetation.**

| Vegetation types | Sample size | Maximum value | Minimum value | Mean value | Standard deviation | Coefficient of variation (%) |
|---|---|---|---|---|---|---|
| Farmland | 89 | 27.8 | 2.6 | 8.5 | 4.2 | 49.6 |
| Forest | 5 | 47.6 | 6.1 | 18.2 | 16.1 | 88.7 |
| Grassland | 14 | 17.8 | 2.2 | 8.1 | 4.4 | 54.3 |

**Table 4 Comparison of model performance for SOC prediction.**

| | RF | | | XGBoost | | | SVM | | |
|---|---|---|---|---|---|---|---|---|---|
| | $R^2$ | RMSE | RPD | $R^2$ | RMSE | RPD | $R^2$ | RMSE | RPD |
| Training | 0.9203 | 1.4275 | 2.4298 | 0.9573 | 1.0887 | 3.1859 | 0.6013 | 2.2822 | 1.5198 |
| Validation | 0.7379 | 8.0292 | 1.0818 | 0.7548 | 7.6792 | 1.1311 | 0.7484 | 8.0274 | 1.0821 |

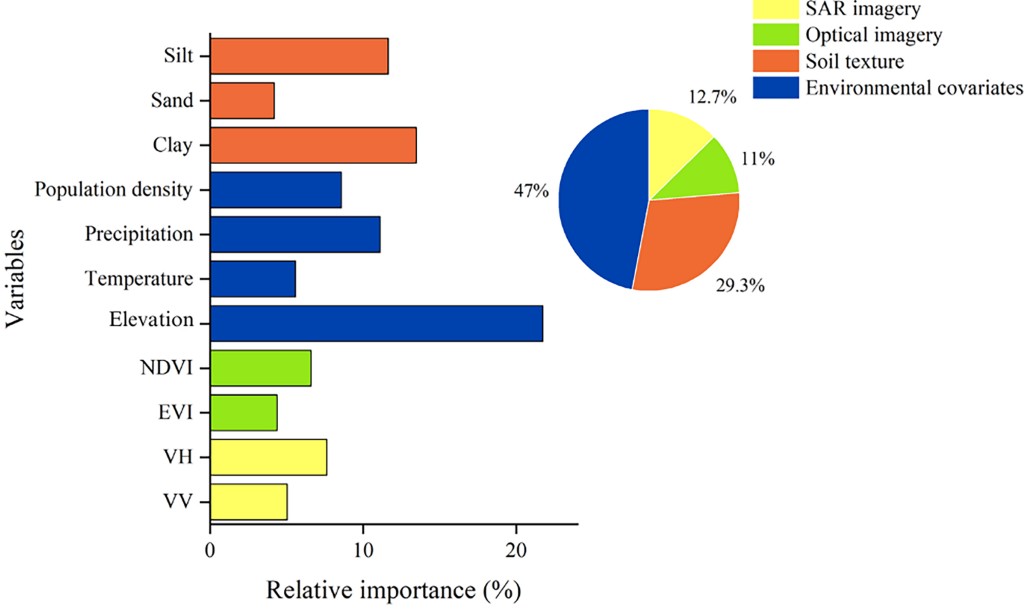

**Figure 2 Relative importance analysis of variables using RF model.**

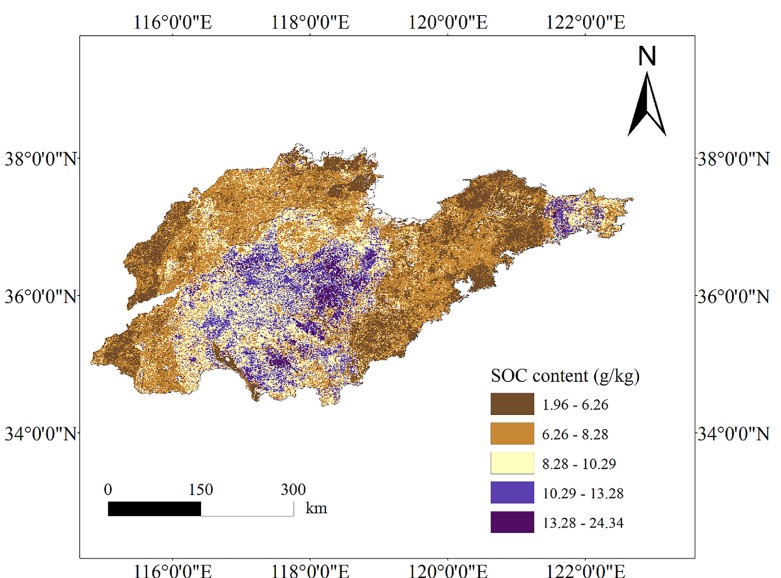

**Figure 3** **SOC content mapping based on the XGBoost model.** Map source credit: DOI: 10.12078/ 2023010103.

SOC content. This suggests that altitude plays a crucial role in predicting SOC content, consistent with previous studies (*Wang, 2021*). Precipitation and temperature had explanatory powers of 11.12% and 5.58%, respectively, indicating a moderate effect on SOC prediction. Precipitation and temperature, as major climatic factors, influence SOC content and its spatial distribution, affecting crop growth and plant net primary productivity (*Wang et al., 2018*). SOC decomposition and accumulation are significantly influenced by climatic hydrothermal conditions. Climate warming accelerates the decomposition of SOC by microorganisms (*Schuur et al., 2015*). Regarding soil type, clay and silt soils exhibit higher explanatory power, amounting to 13.47% and 11.65%, respectively (*Zinn, Lal & Resck, 2005*). Nevertheless, clay content shows relatively weak explanatory power in predicting soil organic matter content, possibly due to the consideration of additional soil physicochemical factors (*Rasmussen et al., 2018*).

The explanatory power of other characterizing variables should also be considered. Population density demonstrates an explanatory power of 8.58%. Regarding the vegetation index, EVI had an explanatory power of 4.38%, while NDVI had 6.6%. In terms of microwave remote sensing, VH had an explanatory power of 7.64%, while VV had 5.05%. These results suggest that besides elevation, precipitation, temperature, soil type, and other environmental factors play significant roles in the spatial distribution of SOC content.

## Spatial distribution analysis of SOC content

The spatial distribution mapping of SOC content was conducted based on the XGBoost simulation results (Fig. 3). To exclude regions with minimal soil coverage, an NDVI mask was applied to the land use classification image. The NDVI mask, established from

literature (*Yang, Di Girolamo & Mazzoni, 2007*), used a threshold value of 0.2 for the NDVI image. Pixels with NDVI values below this threshold were considered to have low vegetation cover and were then masked out from the land use classification image. This process ensured effective exclusion of areas with limited soil presence, leading to a surface SOC content mapping that accurately represents soil-dominant regions.

The study results revealed that in the mountainous areas of central Shandong Province, the highest SOC content was observed in regions with high altitude and steep slopes. These regions, characterized by forest cover, exhibited higher SOC content, consistent with previous research (*Guo et al., 2020*), reaching up to 24.34 g/kg. This can be attributed to the dense vegetation cover, organic matter release by trees during growth (*Zhang et al., 2019*), favorable climatic conditions, and limited anthropogenic impacts, promoting organic matter accumulation and preservation (*Wang, 2019*; *Wiesmeier et al., 2013*). Furthermore, traditional land management practices, like terracing and agroforestry, passed down through generations, aid in soil erosion prevention and organic matter accumulation in the soil (*Chen et al., 2020b*; *Wei et al., 2019*). Cropland areas at low elevations in the western and northern parts of Shandong Province exhibit low SOC content. Common intensive agricultural practices in these areas, such as ploughing and fertilizer application, result in the loss and degradation of soil organic matter (*Wang, Amundson & Niu, 2000*). Adopting sustainable agricultural practices like conservation tillage and organic farming could potentially alleviate the decline in SOC levels in these areas and enhance long-term soil quality (*Martínez-Mena et al., 2020*). The economically developed coastal areas of Shandong Province exhibit lower soil organic matter content attributed to distinct environmental conditions and human activities like urbanization, industrialization, and agricultural development, resulting in land degradation and soil organic matter loss (*Wang, 2019*). Furthermore, the rise in sea levels can cause erosion and degradation of soil carbon reserves (*Haywood et al., 2020*). The proximity to the ocean can result in seawater intrusion, adversely impacting soil quality and the preservation of organic matter (*Morrissey et al., 2014*). Our results align with those of *Dai et al. (2017)*, indicating that surface SOC density distribution follows a pattern of low levels in coastal areas, medium levels in the northwestern plains and eastern hills, and high levels in the mountainous regions of south-central Shandong Province.

Furthermore, in combination with Fig. 4, the study demonstrates variations in how different land use types impact SOC content. Land use in Shandong Province is predominantly cropland, with garden land and woodland following, while grassland, commercial and service land, industrial and mining land are less common. Significantly, woodland and garden land exhibited higher SOC content compared to cropland, aligning with previous studies (*Edmondson et al., 2014*; *Fang et al., 2014*, *2012*) and clarifying the lower SOC content in Shandong Province. This phenomenon may be attributed to the over-utilization of cropland in Shandong Province (*Liu et al., 2005*) and extensive history of repeated ploughing, resulting in decreased SOC content. Optimal practices such as irrigation, fertilizer application, stubble return, and reduced tillage can enhance soil organic carbon storage and agricultural sustainability (*Zhao et al., 2013*).

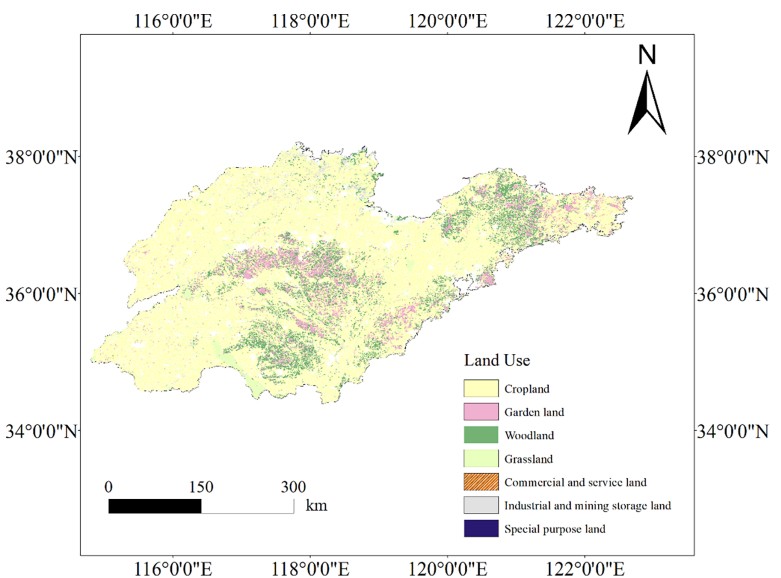

**Figure 4 Land use type of Shandong Province.** Map source credit: DOI: 10.12078/2023010103.

The spatial distribution pattern reveals the dynamics of SOC accumulation and loss across various regions, offering crucial insights for formulating land management policies and executing land conservation strategies.

## CONCLUSIONS

This study used 123 SOC sampling points and RF, SVM and XGBoost machine learning algorithm to construct SOC content retrieval model. Through comparative analysis of the simulation accuracy and stability of the models, the model combination with the best SOC content inversion accuracy was determined. The results showed that: (1) SOC content of Shandong Province was only 8.78 g/kg, and with a high level of variation. (2) Among the three simulation models, XGBoost model obtained the highest predictive accuracy. (3) Among all the influence factors of SOC content, elevation and Clay were identified as the most influence factors, and the explanation reaching to 21.74%, 13.47%. (4) Spatial distribution of SOC content showed that there was a higher SOC content in mountains covered with forest, than plain region covered with croplands, and coast region.

### Funding
The authors received no funding for this work.

### Competing Interests
The authors declare that they have no competing interests.

## Author Contributions

- Qi Chen conceived and designed the experiments, performed the experiments, analyzed the data, prepared figures and/or tables, and approved the final draft.
- Yiting Wang performed the experiments, prepared figures and/or tables, and approved the final draft.
- Xicun Zhu conceived and designed the experiments, prepared figures and/or tables, authored or reviewed drafts of the article, and approved the final draft.

## Data Availability

The raw measurements and code are available in the Supplemental Files.

## Supplemental Information

Supplemental information for this article can be found online at http://dx.doi.org/10.7717/peerj.17836#supplemental-information.

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
