# Peer review of "Soil organic carbon estimation using remote sensing data-driven machine learning"

_PeerJ, doi:10.7717/peerj.17836_

## Round 0.1 · original submission · Major Revisions

You are requesting to revise the manuscript addressing comments of the reviewers.

·

Basic reporting

The work discusses a method for organic soil carbon estimation using remote sensing data-driven machine learning. The work uses random forest, extreme gradient boosting and support vector machine for the estimation. R2 scores and RMSE scores show a promising result.

Experimental design

1. I see Sentinel 1A VV and VH polarization data have been used. This I suppose is a Synthetic Aperture Radar (SAR) data. If it is so, there should be a proper description of the metadata, RSS bands, speckle filters, etc. Further, it is mentioned that NDVI data have been used, however, NDVI itself is not data, it is an index/ratio derived from near-infrared (NIR) and red bands obtained from multispectral optical payloads (sometimes hyperspectral). It is not clear to me what the reviewer is indicating.

2. The machine learning model used in the paper seems like a regression task. For that, the Sentinel 1A data needs to be converted to a time series or simply tabular data with one column a target and the remaining column to train the model. However, the data as described, say NDVI or EVI or perhaps SAR is more close to an image rather than a statistical table. So, how was the data conversion performed? Or if conversion of the data was not done, then how was the model being used to process these data and what were the target variables, and in this cases PSNR values also need to be stated.

3. In "Retrieval model SOC" it is written "expand.grid()", in scientific papers, writings like these are generally discouraged. Programming languages may come and go in some years, but the mathematics/logic/algorithm behind that so called function would stay for centuries to come. So it is advised to write the exact mathematics/logic/algorithm instead of "expand.grid()". Check other places for similar issues.

Validity of the findings

no comment

Reviewer 2 ·

Basic reporting

All comments have been added in detail to the last section.

Experimental design

All comments have been added in detail to the last section.

Validity of the findings

All comments have been added in detail to the last section.

Additional comments

Review Report for "Soil organic carbon estimation using remote sensing data-driven machine learning" in PeerJ

1. Within the scope of the study, soil organic carbon estimation was carried out using remote sensing data with various machine learning models.

2. In the Introduction section; Soil, organic carbon in soil, storage of soil organic carbon and soil organic carbon estimation studies carried out with machine learning in the literature are mentioned at a basic level. In order to emphasize the importance of the study more clearly, it is recommended to add a literature table in this section, consisting of columns such as "dataset used, pros and cons, models used, results". After this, at the end of the introduction section, the difference of the study from the literature and its main contributions to the literature should be added in more descriptive form.

3. In the Materials and methods section; The dataset used within the scope of the study has been mentioned sufficiently. Support Vector Machine, Random Forest and Extreme Gradient Boosting models were preferred as machine learning models. Although there are many different machine learning models that can be used in the literature for soil organic carbon estimation, it should be explained in detail why these three models are preferred and/or why different model trials are not carried out.

4. In the analysis of the results, the required basic metrics such as root mean square error, relative percent difference and coefficient of determination were obtained for each model. When the study is examined in terms of the results obtained and the types of metrics, it can be stated that it is at a sufficient level.

5. In the study, existing machine learning models in the literature for soil organic carbon estimation were used. From this point, it should be stated more clearly what the originality of the study is and whether any improvements have been made in terms of the model.

As a result, the study is important in terms of the problem addressed, but the above-mentioned parts must be clarified in order for the study to fully contribute to the literature.

---

## Round 0.2 · Minor Revisions

The authors are requested to revise the manuscript addressing the final comments from the reviewers.

·

Basic reporting

The paper estimates organic carbon using remote sensing data driven machine learning. The paper is novel and important. The paper uses XGBoost, SVM, RF for the experiment. I have following questions.

1. In abstract, there are too many abbreviations, which need to be avoided. In abstract, what do you mean by DEM? Digital Elevation Model? How is it relevant to the work?

2. Literature survey is not up to the mark. There are several similar papers recently published in this domain. For some example, "Dependence of physiochemical features on marine chlorophyll analysis with learning techniques", "Estimation of chlorophyll-a from oceanographic properties-an indirect approach", "Global marine phytoplankton dynamics analysis with machine learning and reanalyzed remote sensing". A thorough comparison is necessary as all of them are very close to the study that you have performed.

3. More information on data preprocessing is necessary.

4. Equations 1,2,3 aren't novel. Also very old and widely used. Definitions of these might be avoided to save space of the paper. Simply citing them is enough.

5. Results are fine.

Experimental design

See above

Validity of the findings

See above

Additional comments

See above

Reviewer 2 ·

Basic reporting

All comments have been added in detail to the last section.

Experimental design

All comments have been added in detail to the last section.

Validity of the findings

All comments have been added in detail to the last section.

Additional comments

Review Report for PeerJ
(Soil organic carbon estimation using remote sensing data-driven machine learning)

Thanks for the revision. The final version of the study and the revisions made have been examined in detail. Since the subject covered in the study, its originality and its contribution to the literature are sufficient, I recommend that the paper be accepted in its current form. I wish the authors success in their future studies. Best regards.

---

## Round 0.3 · accepted · Accept

The manuscript is ready for production